# In Vitro Evaluation of Eudragit Matrices for Oral Delivery of BCG Vaccine to Animals

**DOI:** 10.3390/pharmaceutics11060270

**Published:** 2019-06-10

**Authors:** Imran Saleem, Allan G. A. Coombes, Mark A. Chambers

**Affiliations:** 1School of Pharmacy and Biomolecular Sciences, Liverpool John Moores University, Byrom Street, Liverpool L3 3AF, UK; I.Saleem@ljmu.ac.uk; 2Pharmacy Australia Centre of Excellence, University of Queensland, School of Pharmacy, 20 Cornwall Street, Woolloongaba, QLD 4102, Australia; allancoombes1@gmail.com; 3Department of Bacteriology, Animal and Plant Health Agency, Woodham Lane, New Haw, Addlestone, Surrey KT15 3NB, UK; 4School of Veterinary Medicine, University of Surrey, VSM Building, Daphne Jackson Rd, Guildford GU2 7AL, UK

**Keywords:** BCG, Eudragit, oral vaccine, tuberculosis, in vitro viability

## Abstract

Bacillus Calmette–Guérin (BCG) vaccine is the only licensed vaccine against tuberculosis (TB) in humans and animals. It is most commonly administered parenterally, but oral delivery is highly advantageous for the immunisation of cattle and wildlife hosts of TB in particular. Since BCG is susceptible to inactivation in the gut, vaccine formulations were prepared from suspensions of Eudragit L100 copolymer powder and BCG in phosphate-buffered saline (PBS), containing Tween^®^ 80, with and without the addition of mannitol or trehalose. Samples were frozen at −20 °C, freeze-dried and the lyophilised powders were compressed to produce BCG–Eudragit matrices. Production of the dried powders resulted in a reduction in BCG viability. Substantial losses in viability occurred at the initial formulation stage and at the stage of powder compaction. Data indicated that the Eudragit matrix protected BCG against simulated gastric fluid (SGF). The matrices remained intact in SGF and dissolved completely in simulated intestinal fluid (SIF) within three hours. The inclusion of mannitol or trehalose in the matrix provided additional protection to BCG during freeze-drying. Control needs to be exercised over BCG aggregation, freeze-drying and powder compaction conditions to minimise physical damage of the bacterial cell wall and maximise the viability of oral BCG vaccines prepared by dry powder compaction.

## 1. Introduction

According to the World Health Organisation (WHO) 2018 Global Tuberculosis (TB) Report, TB remains one of the top 10 causes of death and the leading cause of morbidity from a single infectious agent worldwide [1]. Similarly, livestock TB constitutes a major animal health problem. It has been estimated that >50 million cattle are infected worldwide, costing US$3 billion annually due to reduced cattle productivity, culling and movement and trade restrictions [2]. In addition, the global health burden is increased by the fact that TB in animals is an important zoonosis, causing disease in humans, particularly through the consumption of unpasteurised milk or by aerosol transfer from infected animals. This factor prompted the WHO and allied organisations to publish a joint roadmap for tackling zoonotic TB in 2017 [3]. 

Bacillus Calmette–Guérin (BCG) is a live bacterial vaccine that has been used in humans since 1921. The vaccine was registered for intramuscular administration to badgers in the UK in 2010, but is not currently registered for use in domestic livestock. BCG has been evaluated in numerous trials and in many different species since its creation, and although efficacy can be variable, it is generally considered an effective vaccine when administered parenterally [4]. However, the delivery of BCG to mucosal surfaces through oral or intranasal administration is highly desirable in order to avoid the use of needles for humans and domesticated animals and to simplify the immunisation of wildlife, particularly through the presentation of BCG incorporated in bait [5]. 

The oral route is extensively used for drug delivery, even to exotic species [6], due to its ease of administration and cost effectiveness. Oral vaccines have also been widely investigated, but the difficulty of protecting protein and peptide antigens against degradation in the harsh conditions of the gastrointestinal tract, notably the low pH and the presence of enzymes, remains a major obstacle to progress in this area [7,8,9]. For BCG, data point to the need to protect the live vaccine against degradation in the gut if it is to be effective when administered orally [10]. For example, in experimental studies in humans, doses of oral BCG were administered either with 2% sodium bicarbonate (to neutralise gastric acidity) immediately before [11] or simultaneously with the vaccine [12]. More convincingly, the intra-gastric administration of BCG to brush-tailed possums (the principal wildlife reservoir of TB in New Zealand) was less effective than vaccine administered by the same route in combination with a drug to reduce gastric acidity or when administered intra-duodenally [13,14]. Protection of BCG against degradation in the gut is also the basis of efforts to formulate BCG in alginate microspheres [15] or in a lipid matrix for oral delivery to wildlife [16,17,18]. 

Many pharmaceutical oral dosage forms are enteric-coated with acid-resistant Eudragit^®^ copolymers to protect the stomach mucosa against irritation by drugs, including non-steroidal anti-inflammatory drugs (NSAIDS), or to prevent the degradation of the active component upon exposure to gastric acid or enzyme action [19,20]. The Eudragit^®^ family of copolymers is based on the anionic polymers of methacrylic acid and methacrylates. They contain –COOH as a functional group and dissolve at specific pH values between 5.5 and 7.0, depending upon the grade. Furthermore, these copolymers are available as powders, aqueous dispersions and in solution in inorganic solvents, which renders them highly versatile for the formulation of drug delivery systems. Eudragit has previously been employed for the encapsulation of protein antigens, including colonisation factor antigen I and the F4 fimbriae antigen of *Escherichia coli* (ETEC) [21,22], with the latter being tested as an oral tablet presentation for piglets. Metabolically active lactic acid bacteria have been incorporated in enteric-coated tablets [23] and granules [24], and live-attenuated Ty21a *Salmonella* typhi have been formulated as an oral vaccine in enteric-coated capsules [25]. Recently, a laminate presentation of *Bifidobacterium breve* bacteria dried onto a Eudragit L100 film was shown to protect viable bacterial cells from inactivation by simulated gastric fluid (SGF) [26].

The aim of the present study was to evaluate the potential of a BCG–Eudragit matrix formulation for oral vaccination against TB, with a focus on the potential incorporation into bait for delivery to badgers [5]. We used Eudragit L100 supplied in a powdered form. The polymer dissolves at a pH above 6.0 and has been used as an enteric coating system for oral preparations relevant to this study [21,22,26]. The protective capacity of the matrix formulation towards BCG was evaluated by the exposure of samples to simulated gastric and intestinal fluids (SIF) and the assessment of the viability of BCG was achieved by counting colony-forming units on solid agar medium.

## 2. Materials and Methods 

### 2.1. Materials

BCG Pasteur strain (obtained originally from the Statens Serum Institute, Copenhagen, Denmark) was grown in Middlebrook 7H9 liquid medium plus Albumin Dextrose Catalase (ADC) Supplement (Beckton and Dickinson, UK) to a concentration of approximately 10^8^ colony-forming units (cfu)/mL. Aliquots were prepared and stored at −80 °C for subsequent experiments. Eudragit L100 polymer (ratio of free carboxyl groups to ester groups approximately 1:1, MW 13.5 kDa) was provided by Röhm GmbH (Darmstadt, Germany). Polyoxyethylene-sorbitan monooleate (Tween^®^ 80), mannitol and trehalose were obtained from Sigma-Aldrich (Gillingham, UK). 

### 2.2. Preparation of BCG-Loaded Eudragit^®^ Matrices 

Suspensions of Eudragit L100 powder in phosphate-buffered saline (PBS) (1.9 mL, 18% *w*/*v*) were produced in glass vials with the addition of Tween^®^ 80 (0.1% *w*/*v*) (formulation A). Separate formulations contained, in addition, 0.2% *w*/*v* mannitol (formulation B) or 0.2% *w*/*v* trehalose (formulation C). The Eudragit^®^ aqueous dispersions were acidic and the pH was adjusted to 7.0 using 1 M NaOH prior to sterilisation by autoclaving at 15 psi and 121 °C for 15 min. BCG Pasteur from stock (100 µL, approximately 10^8^ cfu/mL) was added to the sterile suspension and retained at −20 °C for 6 h prior to freeze-drying (60 mbar (6000 Pa) pressure, −50 °C for 24 h (*n* = 3)) using an Edwards Modulyo freeze-drier (Edwards, Burgess Hill, UK). Following freeze-drying, the powders were compressed using a Specac KBr disc compressor (Specac, Orpington, UK) by applying a 4 ton load for 3 min, to produce 13 mm diameter, 2.2 mm thickness BCG-loaded Eudragit matrices (Figure 1). 

### 2.3. BCG Viability during Matrix Formulation

Samples of formulation A were analysed for BCG viability during each stage of BCG–Eudragit matrix production (formulation of suspensions, freezing at −20 °C, freeze-drying and powder compaction). Samples of formulations B and C were analysed for BCG viability during all stages, except after compaction. BCG was extracted from freeze-dried powders and solid matrices by resuspending in 10 mL PBS, pH 7.4 before dilution. Serial dilutions of resuspended samples were made in sterile water containing 0.05% Tween^®^ 80. Aliquots (100 µL) of each dilution were spread onto Middlebrook 7H10 + Oleic ADC (OADC) agar plates (*n* = 3, per dilution) and incubated at 37 °C for 21 days. The number of cfu that resulted was counted for each dilution and converted to an average cfu. 

### 2.4. BCG Viability Following Matrix Incubation in SGF and SIF

Individual BCG–Eudragit matrix samples (*n* = 3) were placed in SGF (10 mL HCl, 0.1 M, pH 1.2) for 2 h at 37 °C in a water bath, removed and dried to constant weight at room temperature and then transferred to SIF (10 mL HEPES, pH 7.4) and retained at 37 °C until they had fully dissolved. BCG was extracted from matrices exposed to SGF by dissolving the matrix in 10 mL of HEPES solution (pH 7, 37 °C). Serial dilutions of BCG extracted from matrices incubated in SGF and SIF were made in sterile water containing 0.05% Tween^®^ 80. Aliquots (100 µL) of each dilution were spread onto Middlebrook 7H10 + OADC agar plates (*n* = 3 per dilution) and incubated at 37 °C for 21 days. The number of cfu that resulted at each dilution was counted and converted to an average cfu. 

### 2.5. Statistical Analysis

Analyses were performed using GraphPad Prism version 7.03 for Windows (GraphPad Software, La Jolla, CA, USA, www.graphpad.com). Losses in BCG viability after each stage of formulation were analysed by two-way ANOVA with Tukey’s multiple comparisons test against the log_10_ transformation of the raw data. The mean viability of BCG at each stage of evaluation was compared for differences between the three formulations using the multiple *t*-test (with each stage of evaluation being analysed separately to avoid assumptions about consistent standard deviations, using the raw data). Significant *p* values were identified using the two-stage linear step-up procedure of Benjamini, Krieger and Yekutieli to control the false discovery rate, with *Q* = 1%. *Q* is the proportion of false discoveries allowed among the discoveries. Differences of *p* < 0.05 were considered significant throughout. 

## 3. Results

The scanning electron micrographs in Figure 2 show the surface appearance of BCG-free matrices produced by compaction of lyophilised powders prepared from dispersions of Eudragit powder in PBS/Tween^®^ 80 (formulation A) plus mannitol (formulation B) or trehalose (formulation C). Numerous pores and fissures are apparent in the lower magnification image of formulation A and at higher magnification for formulation C, compared with formulation B. The typical dense compact structure of BCG-free matrices formed by the compaction of "as received" Eudragit L100 powder alone is shown in Figure 2D. A more "open" morphology would be expected to facilitate the penetration of the BCG-loaded Eudragit matrices by gastric fluids and the deactivation of exposed BCG. However, the presence of PBS salts in the lyophilised powders was considered advantageous, since it provided the possibility of local pH control within the porous matrices during transport through the stomach. Formulation A BCG–Eudragit matrices retained an average of 88% of their initial weight after incubation in SGF for 2 h (216.3 ± SD 8.2 mg reduced to 190.6 ± SD 4.4 mg). When these matrix samples were subsequently transferred from SGF to SIF, complete dissolution occurred within an average of 2.3 h (range, 125 to 150 min). The weight loss of formulations B and C in SGF and the time of dissolution in SIF were not evaluated. However, our investigation of BCG viability following matrix incubation in SGF for 2 h and incubation in SIF until the matrices dissolved revealed that formulations B and C behaved similarly to formulation A, although specific measurements of time and weight were not recorded.

The concentration of the BCG stock solution after thawing was 1.48 × 10^8^ (SD, 0.14 × 10^8^) cfu/mL and 100 µL was incorporated in each of the three types of matrix formulation. Table 1 shows the impact on BCG viability caused by matrix formulation (freezing of BCG–Eudragit co-suspensions, freeze-drying, dry powder compaction), and after the exposure of BCG–Eudragit matrices to SGF and SIF. Figure 3 expresses the same data but illustrates the cumulative reduction in BCG viability for each formulation.

According to two-way ANOVA analysis of the log_10_-transformed data, significant losses in BCG viability were apparently first encountered during the BCG–Eudragit powder suspension stage for all formulations, with reductions of 1.36 to 1.40 log_10_ (Figure 3). No significant additional reduction in viability was caused by freezing the suspensions at −20 °C for 6 h. Further reduction in BCG viability was encountered after freeze-drying for matrix A, but not for matrices B and C, which included mannitol and trehalose, respectively. Thus, upon completion of the freeze-drying stage there was significant retention of BCG viability for matrices B and C compared with A (Table 1, Figure 3). Freeze-dried formulations were compressed to form matrices and then incubated in SGF, followed by SIF. All formulations converged on an incremental loss of BCG viability to around 3.4 × 10^3^ cfu (Table 1), representing a total cumulative average loss in BCG viability of 3.65 log_10_ (range, 3.61–3.74 log_10_). For formulation A, we additionally assessed the impact of compaction on BCG viability. Compaction alone caused a significant, 100-fold reduction in apparent BCG viability compared with the viable concentration of BCG after freeze-drying (Figure 3). It is noteworthy for this formulation that the exposure of BCG–Eudragit matrices to SGF/SIF caused no further reduction in viability, despite their apparently porous morphology (Figure 2), indicating the protective effect of this matrix against low pH conditions. As we did not assess the effect on BCG of compaction of the formulations containing mannitol (B) or trehalose (C), we cannot be sure that the impact of powder compaction on these formulations was equivalent to formulation A. 

## 4. Discussion

BCG-containing Eudragit matrices were formulated using a dry powder compaction approach in an effort to produce an oral BCG vaccine that retains high viability during transit through the gut. The matrices were physically stable for 2 h at low pH (in SGF) and the data indicate that at least the matrix produced from PBS/Tween^®^ 80 did not appreciably permit the low pH medium (SGF) to penetrate, causing loss in BCG viability. Although we did not include a control of BCG incubated in SGF alone, Dobakhti et al. [15] incubated BCG for 1.5 h in SGF at the same pH (1.2) and found the viability was reduced by approximately 1 log_10_. Therefore, our incorporation of BCG in a Eudragit matrix had the desired effect of protecting BCG from the detrimental action of low pH. Although we cannot say with certainty this was the case for the matrices containing mannitol or trehalose, it seems probable. Despite this encouraging result, we encountered a total cumulative average loss in apparent BCG viability from start to finish of 3.65 log_10_. Analysis of the data show that significant losses occurred at the initial formulation stage and at the stage of compaction.

BCG has a tendency to aggregate [27], leading to the apparent effect of reducing the cfu count, which is interpreted as a loss in viability. For this reason, all BCG–Eudragit matrix formulations included Tween^®^ 80, which is commonly added to BCG suspensions prior to culture or vaccine formulation to prevent aggregation [28,29,30]. Despite this, we think aggregation was the most likely explanation for the apparent loss in BCG viability during the initial formulation stage from 1.5 × 10^7^ cfu to an average of 6.2 × 10^5^ cfu across the three formulations; indeed, aggregates of BCG were seen in the initial formulations under the light microscope (data not shown). Bacterial death at this point seems unlikely since the suspensions were adjusted to pH 7.0 before adding BCG. 

A decided disadvantage of the dry powder compaction approach described here for the production of oral BCG vaccine relates to the reduction of BCG viability following powder compaction (evident for formulation A). The application of pressure (e.g., French Press) is recognised as a highly efficient way to disrupt the cell wall of mycobacteria [31]. The pressure normally applied (approximately 100 MPa) is around three times lower than the pressure used to produce the BCG–Eudragit matrices (290 MPa) in the present study. By contrast, a pressure of 5 MPa did not reduce the viability of live bacteria in the tablets produced from hydroxypropyl methylcellulose acetate succinate (HPMCAS) [23]. Thus, the approach of formulating live BCG vaccines by dry powder compaction appears to be severely limited by the sensitivity of the bacteria to compaction. Attempts to reduce the duration and pressure of compaction resulted in matrices that were unstable in SGF (data not shown), suggesting that the inclusion of a binder (e.g., HPMCAS) would be beneficial in future formulations.

We did not investigate the loss of cfu of BCG in formulations B and C during compaction alone, although we show that the cfu of BCG in formulations B and C strongly decreased during two combined steps, i.e., compression and SGF and SIF incubation. Compared to formulation A, formulations B and C contained additional mannitol and trehalose, respectively. However, the ratio of Eudragit to mannitol and trehalose is very high, i.e., 18:0.2 (*w*/*w*). Consequently, we neither expect a major positive or negative effect of mannitol and trehalose on the loss of cfu during compaction, nor dramatic weight loss in SGF.

Gheorghiu et al. [29] reported that freeze-drying results in loss of viability of BCG Pasteur and this was also our observation for matrix A. Loss in BCG viability during freezing, due to ice crystal formation and/or salt crystal growth (arising from the use of PBS as the suspension medium), may have contributed to the rupture of the bacteria cell wall. The addition of mannitol or trehalose provided protection to BCG during freeze-drying, as expected from the literature. Mannitol is widely used as a cryo-protectant to improve the viability of lyophilised formulations and has been considered for the formulation of new, recombinant BCG vaccines for TB [32]. Similarly, the inclusion of trehalose is a common means of stabilising biomacromolecules during drying [33].

The key finding from this study is that formulations of BCG–Eudragit L100 matrices using dry powder compaction resulted in a significant reduction in viability from the start of formulation to final exposure to SGF/SIF, regardless of the formulation used. Although the minimum efficacious dose for oral BCG has not been determined for any species, typical efficacious doses in experimental studies in humans [12] and animals (reviewed in [4]) exceed 10^7^ cfu. It therefore seems unlikely that the present approach of dry powder compaction would produce an efficacious oral vaccine. Nevertheless, the extensive application of Eudragit^®^ for oral drug delivery recommends further studies to exploit the advantages of this material for the production of oral BCG vaccines. Possible strategies include the incorporation of a binder with the dry powders, in order to reduce the compaction forces required for matrix production and thus minimise the physical damage to the bacterial cell walls, and sonication to disperse aggregates of BCG before freezing.

## 5. Conclusions

BCG–Eudragit L100 matrices were produced by dry powder compaction to investigate their potential as oral vaccines for animals. The production of BCG–Eudragit dry powders resulted in significant loss of apparent viability of BCG. Matrix production by powder compaction resulted in a further reduction in BCG viability of around 100-fold. However, the incubation of at least one of the matrices in SGF followed by SIF showed no further reduction in viability, indicating the capacity of the matrices to control SGF ingress and protect the residual live bacteria.

## Figures and Tables

**Figure 1 pharmaceutics-11-00270-f001:**
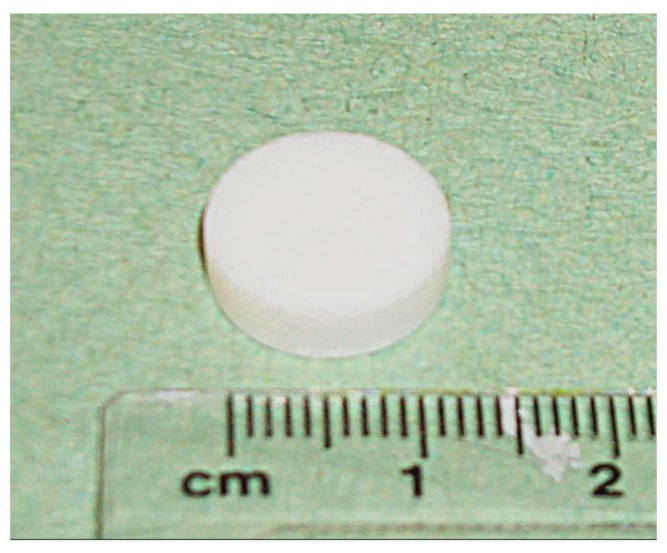
Freeze-dried powders were compressed to produce Bacillus Calmette–Guérin (BCG)-loaded Eudragit matrices with a diameter of 13 mm and a thickness of 2.2 mm.

**Figure 2 pharmaceutics-11-00270-f002:**
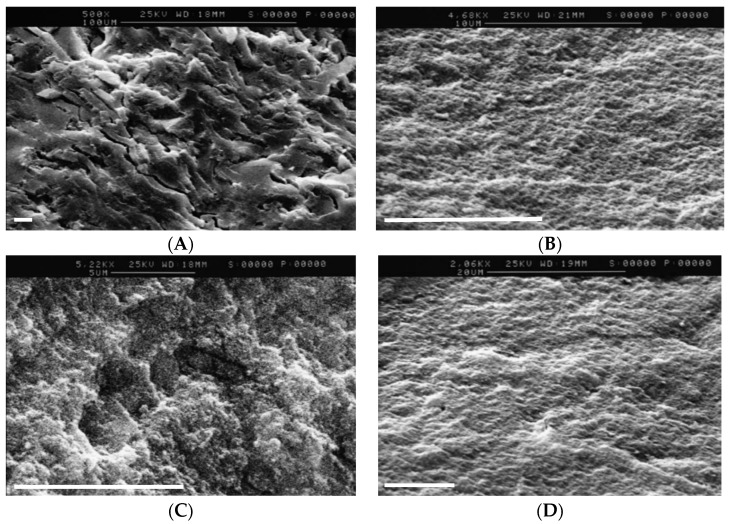
Scanning electron micrographs showing the surface appearance of Eudragit matrices produced by the compaction of lyophilised powders prepared from a dispersion of Eudragit powder in: (**A**) phosphate-buffered saline (PBS)/Tween^®^ 80, (**B**) plus mannitol or (**C**) trehalose; (**D**) or produced by the compaction of "as received" Eudragit L100 powder alone. The size-bar represents 10 µm.

**Figure 3 pharmaceutics-11-00270-f003:**
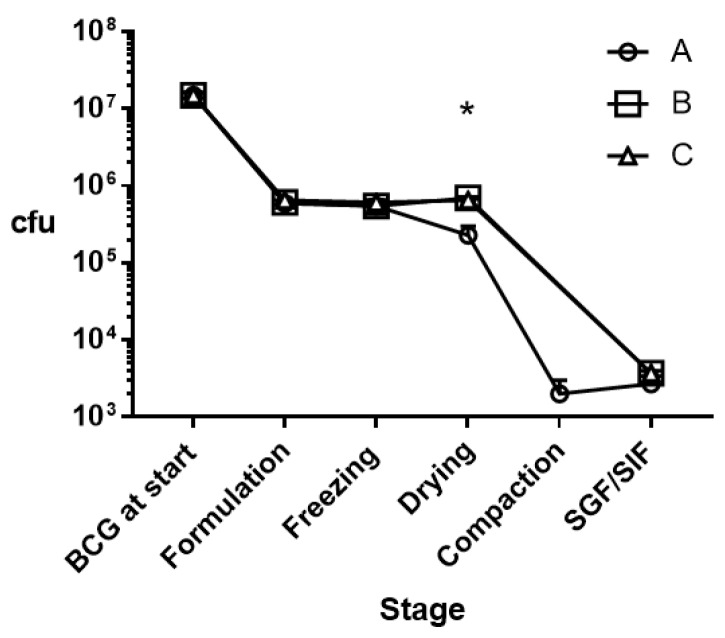
Cumulative reduction in BCG viability (cfu) for each matrix formulation (A, B, C) through each stage. BCG–Eudragit matrix composition: (**A**) Eudragit L100/2 mL PBS/0.9 mL Tween^®^ 80 (0.1% *w*/*v*); (**B**) same as (A) plus mannitol (0.2% *w*/*v*); (**C**) same as (A) plus trehalose (0.2% *w*/*v*). * *p* < 0.05 for pair-wise comparisons of matrices B and C against matrix A.

**Table 1 pharmaceutics-11-00270-t001:** Viability of BCG (colony-forming units (cfu)) at each stage of matrix production (data represent mean ± SD, *n* = 3). For each matrix, 100 µL of BCG stock (1.48 ± 0.14 × 10^8^ cfu/mL) was used, representing a starting BCG quantity of 1.5 × 10^7^ cfu.

Matrix ^1^	Suspension (pH 7)	Freezing (−20 °C/6 h)	Freeze Drying (24 h)	Dry Powder Compaction	Matrix Exposure to SGF and SIF
A	5.9 ± 0.4 × 10^5^	5.5 ± 0.4 × 10^5^	2.3 ± 0.7 × 10^5^	2.0 ± 1.0 × 10^3^	2.7 ± 0.1 × 10^3^
B	6.1 ± 0.4 × 10^5^	5.5 ± 1.1 × 10^5^	7.0 ± 0.2 × 10^5^ *	ND	3.7 ± 0.3 × 10^3^
C	6.5 ± 0.5 × 10^5^	6.1 ± 0.5 × 10^5^	6.5 ± 0.3 × 10^5^ *	ND	3.7 ± 0.2 × 10^3^

^1^ BCG–Eudragit matrix composition: A, Eudragit L100/2 mL PBS/0.9 mL Tween^®^ 80 (0.1% *w*/*v*); B, formulation A plus mannitol (0.2% *w*/*v*); C, formulation A plus trehalose (0.2% *w*/*v*). * *p* < 0.05 for pair-wise comparisons against matrix A.

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
