# Peer review of "In Vitro Evaluation of Eudragit Matrices for Oral Delivery of BCG Vaccine to Animals"

_pharmaceutics, 2019, doi:10.3390/pharmaceutics11060270_

Round 1
Reviewer 1 Report
1. It is stated that formulation A is less resistant to fluid penetration due to a more porous structure (Fig.2). However a water/fluid uptake test, should have enable to differentiate between the uptake capacity pf the three formulation. So the conclusion is rather speculative.The lack of testing on formulation B and C(see 1st paragraph of section 3. Results) prevents from a real understanding of the influence of formulation on matrix disintegration in SGF.
2. Moreover the lack of viability testing on formulation B and C (Table 1 and Fig. 3) after compressions, prevents from concluding about the influence of formulation on cell viability during compression.Therefore on the basis of the presented data, the conclusions appear rather speculative. This should be commented further.
3. In the description of statistical analysis, the meaning of Q =1% might result not clear and should be explained.
Author Response
Please see Word document.

Reviewer 2 Report
The authors presented to formulate BCG vaccine with Eudragit polymers for oral administration. The rationale and design are sound. However, the manuscript lacks key experiments to demonstrate the effectiveness of the designed approach. The detailed suggestions are the following:
Please be more specific in describing the experimental design and the results. For example, on page 4, line 141, the authors should particularize the concentration of the Eudragit polymer.
On page 4, lines 152 & 153, the authors described their observation of retained weight in SGF but complete dissolution in SIF. Although the weight change was recorded in the manuscript. There is no data supporting the dissolution of the matrices in SIF. This is a key experiment in this paper, the authors should present the data. Additionally, the same experiments were not performed for formulations B & C. Why? The authors should perform the same experiments to be consistent.
Table 1, 2 and Figure 3 are different presentations of the same data. I would argue that one set of data do not need 3 ways to present. In addition, these data did not show any advantages of the designed approach. I would suggest the authors add a negative control of BCG (or other previous formulations) treated in SGF/SIF to compare the loss of activity. Alternatively, if possible, the authors can skip the compaction stage for better performance.
Author Response
Please see Word document.

Reviewer 3 Report
In the Introduction section, the authors should have included background of the measure they used to evaluate BCG viability, and why this research is original.
The use of NaOH in the Eudragit aqueous dispersion (line 99) for development of such an oral delivery raised a concern of safety application (toxicity) to patients. How do the authors explain this? Otherwise,this preparation of the oral vaccine is impractical.
Line 123: The authors need to clarify the components of SIF. Why did they use HEPES? Did they follow any guideline or pharmacopeia?
Line 154: Why were Formulation B and C not assessed by that way? How were they assessed?
Line 195-196: The authors had better show real experiments and data rather than an assumption.
Author Response
Please see Word document.

Round 2
Reviewer 2 Report
The authors have made some revisions in the manuscript, but have not provided more experimental evidence to support their conclusions. In particular, the authors have not provided experimental evidence for the weight loss of formulations B and C in SGF and the time of dissolution in SIF. I would suggest the authors provide more evidence.
Author Response
Please see response attached.
